# Making the Case for Autophagy Inhibition as a Therapeutic Strategy in Combination with Androgen-Targeted Therapies in Prostate Cancer

**DOI:** 10.3390/cancers15205029

**Published:** 2023-10-18

**Authors:** Ahmed M. Elshazly, David A. Gewirtz

**Affiliations:** 1Department of Pharmacology and Toxicology, Massey Cancer Center, Virginia Commonwealth University, 401 College St., Richmond, VA 23298, USA; elshazlyam@vcu.edu; 2Department of Pharmacology and Toxicology, Faculty of Pharmacy, Kafrelsheikh University, Kafrelsheikh 33516, Egypt

**Keywords:** autophagy, cytoprotective, abiraterone, bicalutamide, enzalutamide, apalutamide

## Abstract

**Simple Summary:**

This paper is one in a series of articles that investigate the functional forms of autophagy induced in tumor cells in response to various chemotherapeutic modalities, with the overarching goal of determining whether autophagy targeting or modulation could serve as an effective adjuvant therapy. In this review, we focus on androgen-targeted therapies in prostate cancer, including androgen biosynthesis inhibitors and androgen receptor antagonists.

**Abstract:**

Androgen receptor targeting remains the primary therapeutic strategy in prostate cancer, encompassing androgen biosynthesis inhibitors and androgen receptor antagonists. While both androgen-receptor-positive and “castration-resistant” prostate cancer are responsive to these approaches, the development of resistance is an almost inevitable outcome leading to the castration-resistant form of the disease. Given that “cytoprotective” autophagy is considered to be a predominant mechanism of resistance to various chemotherapeutic agents as well as to radiation in the cancer literature, the purpose of this review is to evaluate whether autophagy plays a central role in limiting the utility of androgen deprivation therapies in prostate cancer. Unlike most of our previous reports, where multiple functional forms of autophagy were identified, making it difficult if not impossible to propose autophagy inhibition as a therapeutic strategy, the cytoprotective form of autophagy appears to predominate in the case of androgen deprivation therapies. This opens a potential pathway for improving the outcomes for prostate cancer patients once effective and reliable pharmacological autophagy inhibitors have been developed.

## 1. Introduction

This manuscript is one of a series of papers that were designed to evaluate the role(s) of autophagy in response to various cancer therapeutic modalities. Our previous publications assessed the influence of autophagy in tumor cells on the response/sensitivity to radiation [1], cisplatin [2], microtubule poisons [3], hormonal therapies in estrogen positive breast cancer [4], PARP inhibitors [5], topoisomerase I poisons [6], temozolomide [7], BET family inhibitors [8] and, most recently, BRAF-targeted therapies [9]. This series of papers will ultimately delineate whether there are particular therapeutic modalities where the preclinical data, and where available, clinical trials, support the inclusion of autophagy inhibition or modulation as an adjuvant approach.

## 2. Androgen Receptor Antagonists and Autophagy

Prostate cancer is second only to lung cancer as the most common cancer in men and is considered to be the fifth leading cause of cancer-related deaths among men worldwide [10]. Prostate cancer diagnosis depends on multiple analyses and procedures including prostate-specific antigen (PSA) testing, prostate biopsy and analysis, digital rectal examination, and magnetic resonance imaging, as well as health screening [11]. Generally, prostate cancer can be classified as androgen sensitive or androgen insensitive, an indication of testosterone stimulation and the possible treatment options [12], which include surgery, hormonal therapy, chemotherapy, and radiation therapy. Treatment is based upon the nature of the tumor, PSA level, grade and stage, and the likelihood of disease recurrence [11].

The primary factor promoting prostate tumor cell growth is circulating androgen, particularly dihydrotestosterone (DHT) [13]. Androgen production is initiated with the conversion of cholesterol into pregnenolone, which, upon 3β hydroxysteroid dehydrogenase action, is transformed into progesterone [14,15]. Subsequently, 17α hydroxylase converts progesterone into 17α hydroxyprogesterone, while converting pregnenolone into 17α hydroxypregnenolone. The C17,20 lyase enzyme (CYP17A1) transforms 17α hydroxyprogesterone into androstenedione, as well as converting 17α hydroxypregnenolone into dehydroepiandrosterone [14]. Androstenedione and dehydroepiandrosterone are then converted via enzymatic pathways into testosterone, which is further converted via 5α reductase into dihydrotestosterone [14,16]. These androgens then bind to the androgen receptor(s) on the prostate tumor cells, promoting and maintaining tumor growth.

The androgen receptor (AR) is a nuclear, steroid hormone receptor, for which the gene is located on chromosome Xq11-12. The AR consists of three functional domains: the N-terminal transactivation domain (NTD, exon 1), a DNA-binding domain (DBD, exons 2 and 3), and a C-terminal ligand-binding domain (LBD, exons 5–8) [17,18]. In the absence of androgens, the AR associates with heat shock protein (HSP-90) in a complex that prevents AR degradation and maintains the ligand-binding conformation. DHT binds to the C-terminal LBD of the AR, causing AR dissociation from HSP-90, and resulting in a conformational change and homodimerization of the receptor. Subsequently, the AR translocates to the nucleus binds to DNA via the DBD, recruiting various cofactors and ultimately causing the transcription of androgen-dependent genes that drive tumor growth [19,20,21,22]. 

AR signaling can be driven by genomic amplification and overexpression of AR [23,24], gain of function mutations allowing AR to be persistently activated, alterations in AR transcriptional programs such as enhancer elements [25], co-activators and co-regulators [26], enzyme overexpression [27], and alterations in androgen transport [22,28]. Aside from AR signaling, the tumor may eventually adapt to the low levels of androgen resulting from androgen lowering therapeutic strategies, activating alternative mechanisms for activating AR or by bypassing AR, developing into the castration-resistant form of prostate cancer [29,30].

Androgen deprivation remains the primary approach for the treatment of both androgen-dependent and castration-resistant prostate cancer; the effectiveness of this therapeutic approach in the latter case is indicative of residual dependence of these tumors on androgen stimulation. The principal androgen biosynthesis inhibitor being utilized in clinical settings is abiraterone. Abiraterone selectively blocks androgen biosynthesis via 17α-hydroxylase/C17,20 lyase (CYP17A1), in addition to antagonizing the AR. Regarding androgen receptor antagonists, first-generation agents include bicalutamide, nilutamide, and flutamide while second-generation agents include enzalutamide, apalutamide and darolutamide. These drugs are utilized in the treatment of different stages of prostate cancer; however, as is the case with other antineoplastic agents, the ultimate development of resistance limits their utility. 

In addition to independence from the AR, resistance has been ascribed to genomic alterations, modifications to intracellular signaling pathways, metabolic switches, and possibly autophagy, the topic of this review [27]. Autophagy is a cellular self-digestive machinery that contributes to the maintenance of cellular homeostasis as well as to energy production via controlling degradation of damaged proteins and organelles [3,4,6]. We and others have identified different functional forms of autophagy [4,31,32]. However, as indicated in the studies presented below, the cytoprotective form of autophagy appears to be the exclusive form observed in response to androgen deprivation therapies in preclinical models of prostate cancer (Figure 1), suggesting that autophagy modulation or targeting could be an effective strategy to increase the effectiveness of androgen receptor antagonists [33,34].

## 3. Abiraterone and Autophagy

Abiraterone is an inhibitor of androgen biosynthesis that is commonly utilized in combination with chemical castration and prednisone for metastatic castration-resistant and metastatic castration-sensitive prostate cancer [35,36]. In addition to the antagonism of AR, which mediates the actions of abiraterone in CYP17A1 low-expressing cells (ex: LNCap) [37,38], the primary mode of action for abiraterone is the selective inhibition of androgen biosynthesis via a blockade to 17α-hydroxylase/C17,20 lyase (CYP17A1) [39]. By blocking the action of CYP17A1, abiraterone suppresses androgen production not only in the testes but also in other androgen-producing tissues including the adrenal glands and the prostate tumor [40]. Although abiraterone demonstrates a robust antineoplastic activity, variable responses in prostate cancer patients and the development of resistance limits its clinical efficacy [41].

Mortezavi et al. [42] studied autophagy targeting as a potential strategy in combination with abiraterone in prostate cancer, using the standard LNCaP, DU145, and PC3 cell lines. Here, it should be noted that these are human-derived cell lines, where the LNCaP cells express the AR and are androgen dependent, whereas the DU145 and PC 3 cells do not express the AR and are androgen independent. What must be emphasized additionally is that the DU145 and PC3 cells do not actually reflect the clinical castration-resistant phenotype, wherein androgen deprivation is an effective strategy, at least initially. Furthermore, being human derived, these tumor cells cannot generally be studied in immune-proficient animals, except in the case where immune deficient mice have been humanized [43].

Abiraterone treatment, at concentrations of 5, 10, and 15 µM for 1 and 4 days, reduced the proliferation of the LNCaP cells in a concentration-dependent manner, without affecting the DU145 and PC3 cell lines, consistent with abiraterone effectiveness requiring androgen sensitivity. Of note, the concentrations being utilized are not reflective of the clinical concentration as the C_max_ reported in patients’ plasma is approximately between (3.3 µM) 1173ng/mL and (0.64 µM) 226 ng/mL and may be lower [44,45,46]. Abiraterone induced autophagy in the LNCaP cells, as assessed by elevated levels of ATG5 and Beclin 1 proteins, together with increasing the conversion of LC3 I to LC3 II [47]. The induction of autophagy was further validated by reduced p62/SQSTM1 protein expression by Western blot analysis [48]. Autophagosome formation was also monitored by AUTOdot as well as by immunofluorescence, where LNCaP cells treated with abiraterone exhibited cytoplasmic elevation in ATG5 expression and a punctuated pattern for LC3, confirming the accumulation of autophagosomes [42].

The role of the autophagy induced in the LNCaP cells upon abiraterone treatment was evaluated utilizing pharmacological and genetic inhibition of autophagic flux (i.e., autophagy completion with degradation of the autophagosomal substrates). Combining abiraterone with either of the pharmacologic autophagy inhibitors, CQ or 3-MA, increased the rate of cell death, reducing the viability of LNCaP cells as compared to either treatment alone. Increased cell death caused by the combination treatment was accompanied by increased levels of apoptosis as assessed by annexin V/PI staining, all of which is consistent with a cytoprotective role of autophagy. This cytoprotective function of autophagy was further confirmed using ATG5-directed siRNA in the LNCaP cells. As was the case with pharmacologic inhibition of autophagy, ATG5-deficient LNCaP cells exhibited a significantly increased sensitivity towards the cytotoxicity of the abiraterone treatment.

Ma et al. [49] also studied autophagy inhibition in combination with abiraterone in the PC3 and LNCaP cell lines. Comparing the androgen-sensitive LNCaP cells and the castration-resistant PC3 cells, the PC3 cells were found to have higher levels of basal autophagy based on TEM-detected autophagic vacuoles, and increased ATG5, LC3II, and Beclin1 levels. Abiraterone treatment at 10 µM for 48 and 72 h reduced the viability of both the PC3 and LNCaP cells, as shown by a CCK-8 assay, together with the promotion of apoptosis, based on the expression of cleaved caspase-3 and a decrease in the anti-apoptotic protein, BCL-2. Moreover, abiraterone caused G2/M cell cycle arrest in both cell lines after treatment for 48 h. Unexpectedly, and in contrast to the findings by Mortezavi et al. [42], abiraterone treatment significantly *inhibited* autophagic flux in both cell lines, as confirmed by reduced levels of TEM-detected autophagic vacuoles, and reduced levels of ATG5, Beclin1, and LC3II, which was further confirmed by immunofluorescence. However, upon combining abiraterone with 3-MA, an enhanced reduction in cell viability and the promotion of apoptosis, as well as G2/M arrest, were evident. The inhibition of autophagy in response to the combination of 3-MA and abiraterone was confirmed by a decrease in autophagic vacuoles, reduced ATG5, LC3II, and Beclin1 levels, and a reduced positive staining of LC3II in the two cell lines. These data suggested that castration-resistant cells have an intrinsic protective autophagy that is inhibited by abiraterone. This is in contrast to the studies by Mortezavi et al. [42], which showed that abiraterone induced autophagy in LNCaP cells. However, additional experiments that include the genetic inhibition of autophagy in these cell lines would be necessary to further confirm the role of the autophagic machinery [50]. 

Recently, Feng et al. [51] studied Qi Ling decoction (QLD), a Chinese herb, in combination with abiraterone using PC-3 and DU145 cells that were developed to be abiraterone resistant. Using flow cytometry and CCK-8 assays, QLD in combination with abiraterone resulted in decreased cell survival together with robust apoptosis in the resistant cell lines as compared to each drug alone. Importantly, as shown by a GFP-LC3 fluorescence assay, the autophagic levels were higher in abiraterone-treated cells as compared to either QLD alone or QLD in combination with abiraterone. These results were further confirmed by studies demonstrating that abiraterone induced a significant elevation in the LC3II/LC3I ratio as well as in Beclin1 protein levels. QLD addition to abiraterone-treated cells partially suppressed the abiraterone effect on LC3-II/LC3-I ratios and Beclin1 expression, suggesting that QLD reduced the autophagic flux induced by abiraterone in both the PC3 abiraterone-resistant and DU145 abiraterone-resistant cells. The cytotoxicity induced by abiraterone in combination with QLD was mirrored in vivo using PC3 abiraterone-resistant or DU145 abiraterone-resistant tumor-bearing mice models, where QLD combined with abiraterone produced a significant tumor inhibition activity as compared to each drug alone. These results strongly suggest a cytoprotective role of abiraterone-induced autophagy, but still require further verification using pharmacologic as well as genetic autophagy inhibition studies [50].

Collectively, only limited information relating to abiraterone and its possible relationship with autophagy is available in the literature, with contradictory findings as to whether autophagy is induced or suppressed in response to abiraterone. Furthermore, it is critical to highlight the contradictory results regarding whether abiraterone has an effect in the AR-deficient PC3 and DU145 cell lines. However, based on the studies described above, autophagy seems to play a cytoprotective role in prostate cancer, regardless of whether abiraterone itself induces or suppresses autophagy.

## 4. Bicalutamide and Autophagy

Bicalutamide is a potent, first-generation, nonsteroidal antiandrogen with a long plasma half-life, consistent with its administration once daily [52]. Bicalutamide demonstrates significant activity in prostate cancer, with a tolerable safety profile, offering a better choice than flutamide. Bicalutamide represents a valid first choice for antiandrogen therapy in combination with chemical castration, or a GnRH agonist for the treatment of patients with advanced prostate cancer [52,53]. A number of studies have investigated the possible targeting of autophagy either to increase the effectiveness of bicalutamide or to overcome the development of resistance. For example, Nguyen et al. [33] reported that bicalutamide induced autophagy in both LNCaP cells and the C4-2B, AR-positive cell line derived from LNCaP cells [54], as shown by the transition of LC3I to LC3II. Mechanistically, AMPK activation, an upstream promoter of autophagy, was reported to be significantly increased in cells treated with bicalutamide.

Consistent with the findings of Nguyen et al. [33], Boutin et al. [55] showed that bicalutamide treatment or androgen deprivation induced autophagy in LNCaP cells, as indicated by GFP-LC3 accumulation into vacuoles. The induction of autophagy was further confirmed by the detection of autophagosomes and autophagolysosomes by TEM and increased fluorescence with a Cyto-ID autophagy detection assay, as well as by p62/SQSTM1 degradation. Androgen deprivation or bicalutamide treatment in an AR-devoid U-145 cell line failed to promote autophagy, suggesting that autophagy induction is directly associated with AR signaling inhibition.

To determine the role of bicalutamide-induced autophagy, the ATG5 gene was targeted using siRNA in LNCaP cells. ATG5 depletion increased apoptosis upon bicalutamide treatment, suggesting a cytoprotective role of autophagy. The cytoprotective role of autophagy was confirmed pharmacologically in that CQ in combination with either androgen deprivation or bicalutamide increased the level of apoptosis, as assessed by sub-G1 detection, and the plasma membrane permeabilization measurement. Similar results were obtained using concanamycin A, an autophagy inhibitor that blocks vacuolar H^+^-ATPase [56], which enhanced the toxicity of the androgen deprivation and bicalutamide treatment.

Mechanistically, androgen deprivation or bicalutamide treatment reduced the phosphorylation of Akt^S473^, the upstream kinase of mTOR, as well as of p70S6k^T389^ and 4EBP-1, the substrates of mTOR. Interestingly, autophagy induction was shown to be associated with AR/PI3-K interaction in LNCaP cells, based on the association (by co-immunoprecipitation) between AR and p85, the regulatory sub-unit of class Ia PI3-K. Therefore, it was proposed that androgen ablation, by decreasing AR expression, increased the cytosolic free p85 that in turn inhibits PI3-K activity, which, in turn, further inhibits the PI3-K/Akt/mTOR pathway, a master regulator of multiple forms of autophagy [57].

Collectively, these data support a cytoprotective role of autophagy in response to bicalutamide treatment in AR-dependent prostate tumor cells as AR-devoid cell lines did not demonstrate an autophagic response [55,58].

## 5. Enzalutamide and Autophagy

Enzalutamide is a second-generation, nonsteroidal AR inhibitor that is widely used to treat prostate cancer, especially the metastatic castration-resistant form of the disease [59,60]. Enzalutamide belongs to the class of direct androgen receptor inhibitors, such as apalutamide [10]. Enzalutamide targets the AR pathway at multiple stages, specifically blocking AR binding to androgen, blocking AR transcriptional activity by preventing AR from translocating to the nucleus, inhibiting DNA transactivation via binding to DNA and recruiting cofactors [59,60], and consequently suppressing the expression of androgen-responsive genes [59,61]. The multiple effects exerted by enzalutamide on AR pathways are considered the primary basis for its superior clinical efficacy over flutamide, bicalutamide, or other antiandrogen drugs [62,63,64]. However, due to prostate cancer heterogeneity, the response to enzalutamide treatment varies between patients [65]. In addition, the development of resistance is widely recognized as a major drawback to therapy efficacy. 

Nguyen et al. [33] investigated the potential utilization of autophagy targeting to overcome enzalutamide resistance using LNCaP, C4-2B, enzalutamide-resistant C4-2B, CWR22Rv1, and PC-3 prostate cancer cell lines. Enzalutamide, at 10 μM for 48 h, induced autophagy in both androgen-responsive LNCaP and androgen-insensitive, AR-positive, CWR22Rv1 cells, as evidenced by the emergence of bright punctate fluorescence in autophagosomes using an LC3-eGFP assay. The induction of autophagy was further confirmed by an increase in the ratio of LC3II/LC3I as well as the upregulation of ATG5. The increase in autophagosome formation was further validated by Flow cytometry. Interestingly, enzalutamide-resistant C4-2B cells were shown to have a high level of basal autophagy, based on acridine orange staining and an increased LC3II/LC3I ratio, consistent with the possibility that autophagy could be contributing to enzalutamide resistance. Furthermore, the genes involved in autophagosome formation, ULK1, ATG12, ATG16L2, DRAM1, and DRAM2, were found to be upregulated in the resistant cell lines. In addition, several mTOR signaling genes were differentially downregulated in the resistant cells when compared with the parental cells, where suppression of mTOR is known to promote autophagy [57]. Enzalutamide was unable to induce autophagy in AR-negative PC3 cells, as shown by the unaltered levels of LC3I and LC3II, again suggesting that autophagy occurs in parallel with AR inhibition. 

To determine the role of the autophagy induced by enzalutamide, autophagy was inhibited using CMI, an FDA-approved drug to treat depression, which has been shown to be a potent inhibitor of autophagy with minimal toxicity both in vitro and in vivo [66,67]. The combination of enzalutamide and CMI reduced the colony-forming ability of C4-2B cells as compared to each drug alone. Furthermore, in an in vivo model of SCID mice orthotopically implanted with enzalutamide-resistant cells, the combination of enzalutamide with CMI produced a more pronounced reduction in tumor size as compared to each drug alone. The replacement of CMI with metformin as an autophagy modulator [68] resulted in similar results in vivo, with a significantly greater reduction in tumor size than each drug alone. These results strongly suggest a cytoprotective role of enzalutamide-induced autophagy in this model; however, genetic inhibition of autophagy studies would be needed to further confirm this proposed cytoprotective function of autophagy [50].

Mechanistically, Nguyen et al. [33] reported that AMPK activation, which is association with the promotion of autophagy [69], was significantly increased in LNCaP and C4-2B cells treated with enzalutamide. Similarly, enzalutamide-resistant C4-2B cells showed an upregulation of AMPK, suggesting a primary role for AMPK in the induced autophagic machinery. This role of AMPK was further confirmed using siRNA suppression of AMPK in C4-2B cells, where enzalutamide treatment then failed to induce autophagy, as evidenced by LC3-eGFP analysis. Upon AMPK knockdown using siRNA in LNCaP and C4-2B cells, enzalutamide increased cell death. Furthermore, AMPK knockdown sensitized enzalutamide-resistant C4-2B cells, as shown by enhanced cell death, suggesting the crucial role of AMPK in the development of resistance that is maintained by autophagy.

The AMPK pathway directly interacts with the TSC2/Raptor/mTOR complex to inhibit mTOR/S6K/4EBP signaling and to subsequently induce autophagy [70,71,72]. Nguyen et al. [33] showed that enzalutamide treatment was coupled with AMPK activation and increased the phosphorylation of Raptor, resulting in increased LC3-I to LC3-II conversion, whereas p-AKT remained unaffected. However, upon AMPK knockdown, enzalutamide treatment did not affect the phosphorylation of Raptor, while it did reduce ATG5 expression and reduced the conversion of LC3-I to LC3-II. Therefore, Nguyen et al. [33] proposed the potential interaction between AMPK activation and the suppression of mTOR via the phosphorylation of Raptor upon the induction of enzalutamide-mediated autophagy. This hypothesis was further tested by a co-immunoprecipitation analysis, where, upon knockdown of the mTOR complex, phospho-Raptor appeared when AMPK was activated by enzalutamide treatment; conversely phospho-Raptor was undetectable when AMPK expression was knocked down.

Although there are limited publications available assessing the relationship between autophagy and enzalutamide, as is the case with bicalutamide, the available studies collectively strongly suggest that enzalutamide induces autophagy in prostate tumor models. Furthermore, the autophagy induced has a direct relationship to the development of enzalutamide resistance, suggesting a cytoprotective role of autophagy. As is the case with bicalutamide, it is important to highlight what Nguyen et al. [33] reported regarding the relationship between AR inhibition and autophagy induction, with no autophagy induced in the AR-devoid PC-3 cell line.

## 6. Apalutamide and Autophagy

Apalutamide is a novel, potent, second-generation AR antagonist that binds the AR with 7- to 10-fold greater affinity than bicalutamide [73]. Apalutamide exerts its action by directly inhibiting the AR at the ligand-binding domain, thereby suppressing AR nuclear translocation, DNA binding, and AR-mediated transcription [74]. Apalutamide showed superior efficacy in phase I and II trials in patients with non-metastatic castration-resistant prostate cancer by Rathkopf et al. [75] and Smith et al. [76], respectively. These clinical trials, together with the data from the SPARTAN trial by Smith et al. [77], in which apalutamide showed a significant antitumor activity, led to the approval of apalutamide for treating non-metastatic castration-resistant prostate cancer. However, some patients do not respond to this therapy, and eventually become resistant. Recently, Eberli et al. [78,79] studied the potential targeting of autophagy to increase the effectiveness of and possibly overcome resistance to apalutamide both in vitro and in vivo.

Eberli et al. [78] investigated the relationship between autophagy inhibition and apalutamide in prostate cancer using the LNCaP cell line. Apalutamide reduced proliferation of the LNCaP cells in a dose-dependent manner. Apalutamide also induced autophagic flux, as confirmed by increased ATG5 and Beclin-1 expression, as well as the punctuated pattern for LC3, using immunofluorescence analysis. In addition to the reported increase in ATG5 and Beclin-1, reduced p62/SQSTM1 levels as well as the conversion of cytosolic LC3-I to membrane-bound LC3-II were detected. The induction of autophagy was further validated using AUTOdot fluorescence staining, in which the characteristic features of autophagy, such as increased cell size and autophagic vacuoles, were revealed upon apalutamide treatment.

The pharmacological inhibition of autophagy using CQ or 3-MA in combination with apalutamide significantly reduced cell proliferation as compared to each treatment alone, suggesting a cytoprotective role of autophagy in this experimental system. The enhanced efficacy of autophagy inhibitors combined with apalutamide was further demonstrable by a reduction in cell viability, using ethidium bromide and detected by flow cytometry, as well as elevated levels of apoptosis, as shown by an annexin V assay. The depletion of ATG5 utilizing a siRNA strategy also significantly increased cell death and increased apoptosis.

In vivo, Eberli et al. [79] studied the possibility of autophagy targeting in combination with apalutamide, using LNCaP-injected castrated nude mice models. The apalutamide-treated mice showed signs of autophagy induction, as evidenced by the increased expression of ATG5 and a punctuated pattern for LC3 by immunofluorescence. The induction of autophagy was further confirmed based on increasing levels of ATG5 and Beclin-1 detected by immunoblotting. Furthermore, treatment with the combination of apalutamide and CQ significantly reduced tumor Ki-67 fluorescence intensity and tumor weights and increased cleaved caspase-3 levels as compared to each drug alone. 

These results from Eberli et al. [78,79] strongly suggest the cytoprotective autophagy is being induced by apalutamide in LNCaP cells either in vitro, or after being implanted in mice models, highlighting the possible targeting of autophagy to increase the effectiveness of apalutamide. Although promising, a major limitation for these studies, as well as that of Boutin et al. [55], is the utilization of a single cell line, LNCaP cells, which may limit the generalization of these results to other prostate cancer cell lines and the clinical situation.

## 7. Conclusions

Androgen-targeted therapies are currently utilized in clinical settings in the treatment of various stages of prostate cancer; however, as is the case with other chemotherapeutic agents [3,80], the development of resistance constrains their clinical efficacy. Whereas four different functions of autophagy have been identified in the scientific literature [32], with the recent advent of additional subforms such as mitophagy [81], ER-phagy [82], pexophagy [83], and aggrephagy [84], the literature relating to androgen deprivation strategies in prostate cancer appears to consistently demonstrate the promotion of the cytoprotective form that could potentially be suppressed for therapeutic benefit. In this context, HCQ and CQ are widely used in pre-clinical and clinical trials as autophagy inhibitors to increase the effectiveness of various chemotherapeutic modalities [3,7]. However, it remains uncertain whether the doses of these agents that are tolerable in patients actually suppress autophagy in the tumors. 

As summarized in Table 1, the data relating to whether abiraterone induces or suppress autophagy in prostate cancer cells is somewhat inconsistent. There are also somewhat contradictory findings reported regarding whether abiraterone has an effect in PC3 and DU145 cell lines. Mortezavi et al. [41] and Fragni et al. [85] reported that abiraterone did not affect the viability of PC3 and DU145 cells; in contrast, both Ma et al. [49] and Giatromanolaki et al. [86] showed that abiraterone treatment suppressed the viability and/or inhibited the growth of PC3 cells, with the latter suggesting that the attenuation of AR signaling is not the only rationale to explain the abiraterone anticancer activity. However, bicalutamide, enzalutamide, and apalutamide induced autophagy in different tumor models, while one study by Boutin et al. [55] showed that flutamide induced autophagy in LNCaP cells. No information is currently available as to whether nilutamide or darolutamide promote autophagy. 

Although the evidence for the cytoprotective autophagy associated with androgen deprivation needs to be further validated via the pharmacologic and genetic inhibition of autophagy in different cell lines, and importantly, in vivo, using different tumor models [50] and physiologically relevant concentrations (in contrast to some studies mentioned earlier that used supraphysiological concentrations of AR antagonists), autophagy inhibition in prostate cancer likely has the potential to increase the efficacy of androgen antagonists. These studies highlight the possible translation of autophagy inhibition as a strategy to increase the clinical response to androgen deprivation strategies in prostate cancer. 

## Figures and Tables

**Figure 1 cancers-15-05029-f001:**
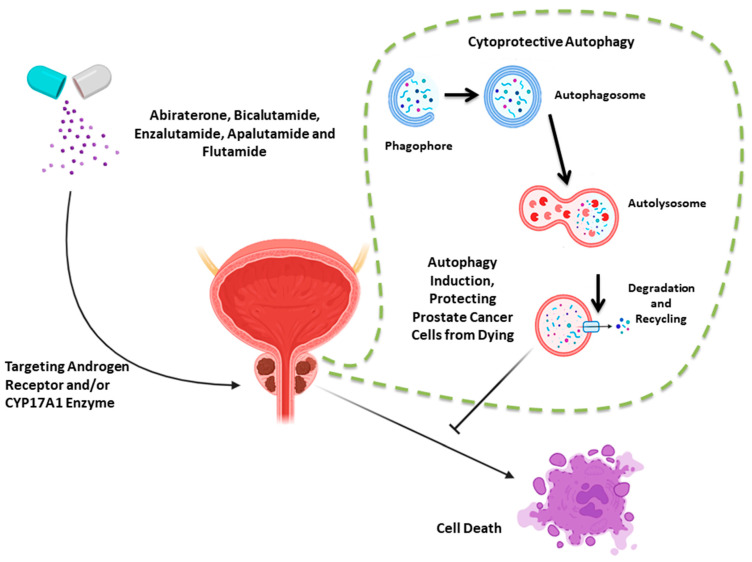
Autophagy induced in response to androgen-targeted therapies. Androgen-targeted therapies (abiraterone, bicalutamide, enzalutamide, apalutamide, and flutamide) exert their effects on prostate cancer cells via androgen receptor blocking and/or targeting the CYP17A1 enzyme. These agents, however, induce a cytoprotective form of the autophagic machinery (autophagy steps: formation of phagophore, then autophagosomes, fusion between autophagosomes and lysosomes, forming autolysosomes in which the cargo is degraded), which protect the prostate cancer cells from undergoing cell death.

**Table 1 cancers-15-05029-t001:** Autophagy roles in response to androgen-targeted therapies.

Androgen Receptor Blocker	Tumor/Cell Type	Autophagy Induction	Role of Autophagy	References
Abiraterone	LNCaP, DU145, and PC3 cell lines	Autophagy induced in LNCaP cells	cytoprotective	[42]
Abiraterone	PC3 and LNCaP cells	Autophagy supressed	-	[49]
Abiraterone	PC3 abiraterone-resistant and DU145 abiraterone-resistant In vivo, using PC3 abiraterone-resistant or DU145 abiraterone- resistant tumor-bearing mice models	Autophagy induced	cytoprotective	[51]
Bicalutamide	LNCaP and C4-2B prostate cancer cell lines	Autophagy induced	-	[33]
Bicalutamide	LNCaP and AR-devoid U-145 cell line	Autophagy induced in LNCaP cells but not in U-145 cells.	cytoprotective	[55]
Enzalutamide	LNCaP, C4-2B, enzalutamide-resistant C4-2B, CWR22Rv1, and PC-3 prostate cancer cell lines In vivo model; SCID mice and orthotopically implanted enzalutamide-resistant cells into the prostate	Autophagy induced but not in AR-devoid PC-3 cells	cytoprotective	[33]
Apalutamide	LNCaP cell line In vivo, using LNCaP-injected castrated nude mice models	Autophagy induced	cytoprotective	[78,79]
Flutamide	LNCaP cell line	Autophagy induced	-	[55]

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
