# Peer review of "Making the Case for Autophagy Inhibition as a Therapeutic Strategy in Combination with Androgen-Targeted Therapies in Prostate Cancer"

_cancers, 2023, doi:10.3390/cancers15205029_

Round 1

Reviewer 1 Report

Drs. Elshazly and Gewirtz have compiled a review on the role of autophagy on the efficacy of prostate cancer therapeutics that target the androgen receptor (AR) axis. The review points out to evidence that suggest that the efficacy of androgen synthesizing enzyme inhibitor abiraterone and AR antagonists such as bicalutamide, enzalutamide, and apalutamide is weakened by their indirect and unintended increase in autophagy. The authors also point out that in preclinical models a combination of AR axis inhibitors and an autophagy inhibitor prevent/reverse any resistance, suggesting that autophagy is the primary cause of resistance to treatment in prostate cancer.

Although the role of autophagy in prostate cancer is important, the effect of various drugs on autophagy is similar making the description of mechanism of various drugs redundant. The authors are requested to expand beyond the AR signaling axis to other drug classes and make the review broader than the AR signaling axis. Otherwise, various sections represented by different drugs will be identical and will lack any depth. Are there any evidences for the cross talk between autophagy and AR splice variants. Also, kindly address the following inconsistencies.

1.      Lines 84-86 : Please revise the sentence “This appears to be ….”. It is currently well established that androgen signaling axis plays a pivotal role in CRPC. Earlier the disease was called androgen-independent PC, the description of which was later changed to CRPC.

2.      Lines 107-109 : Abiraterone’s primary mechanism of action is cyp17A1 inhibition. AR antagonism is only a secondary mechanism. Please shuffle the sentence to make cyp17A1 as the primary MOA.

3.      Lines 152-156 : It is unclear why abiraterone is inhibiting PC3 cell proliferation and inducing autophagy. Considering that PC3 cells are AR independent, there should have been no effect. Please consider removing this mention.

4.      Lines 169-186 : Same issue as the above point.

5.      Since apalutamide and enzalutamide belong to the same scaffold and are closely related, it would be better to combine the two into one section. If data is available for darolutamide, it may be included.

Reviewer 2 Report

There is a whole section about in vitro treatment of compounds with abiraterone, but the major target of abiraterone, CYP17A1, isn't expressed in prostate cells.  It's expressed in adrenal glands where it provides most of the function.  CYP17A1 isn't even the major target when prostate cancer cells upregulate androgen synthesis pathways.  The authors should comment on these findings.  High, supraphysiological concentrations than those seen in vivo in patients utilized in in vitro culture, as described by the authors, can work as weak anti-androgens. Particularly in LNCaP cells which have a T878A AR mutation that allows for promiscuous ligand binding. The authors should comment on the nature of these experiments.

There is some "clunky" word usage.  More precise language should be utilized, for example. when talking about things like AR-inhibitors, use that, or AR-antagonists or anti-androgenic compounds instead of  "androgen blockers."

Round 2

Reviewer 1 Report

None.

Reviewer 2 Report

Thank you for addressing my comments, but I still don't understand how treating cancer cells with high levels of abiraterone in culture produces an on-target effect through androgen synthesis inhibition.  The authors suggest that this is the mechanism of action, but others have shown that high concentrations in culture media produce anti-tumor effects through direct AR-inhibition.  Especially given that CYP17A is not expressed in LNCaP cells.  Can the authors add this to the abiraterone section? Here is a citation: https://www.ncbi.nlm.nih.gov/pmc/articles/PMC5451248/
